# Monoclonal Antibodies against Calcitonin Gene-Related Peptide for Migraine Prophylaxis: A Systematic Review of Real-World Data

**DOI:** 10.3390/cells12010143

**Published:** 2022-12-29

**Authors:** Antun R. Pavelic, Christian Wöber, Franz Riederer, Karin Zebenholzer

**Affiliations:** 1Department of Neurology, Hietzing Hospital, 1130 Vienna, Austria; 2Department of Neurology, Medical University of Vienna, 1090 Vienna, Austria; 3Comprehensive Center for Clinical Neurosciences & Mental Health, Medical University of Vienna, 1090 Vienna, Austria; 4Faculty of Medicine, University of Zurich, 8032 Zurich, Switzerland

**Keywords:** real-world, erenumab, galcanezumab, fremanezumab, eptinezumab, pharmacoepidemiology, effectiveness, tolerability, safety, treatment pause, switching

## Abstract

Objective: To perform a systematic review of real-world outcomes for anti-CGRP-mAbs. Methods: Following the PRISMA guidelines, we searched PubMed for real-world data of erenumab, galcanezumab, fremanezumab, or eptinezumab in patients with migraines. Results: We identified 134 publications (89 retrospective), comprising 10 pharmaco-epidemiologic and 83 clinic-based studies, 38 case reports, and 3 other articles. None of the clinic-based studies provided follow-up data over more than one year in more than 200 patients. Findings suggest that there are reductions in health insurance claims and days with sick-leave as well as better treatment adherence with anti-CGRP-mAbs. Effectiveness, reported in 77 clinic-based studies, was comparable to randomized controlled trials. A treatment pause was associated with an increase in migraine frequency, and switching to another antibody resulted in a better response in some of the patients. Adverse events and safety issues were addressed in 86 papers, including 24 single case reports. Conclusion: Real-world data on anti-CGRP-mAbs are limited by retrospective data collection, small patient numbers, and short follow-up periods. The majority of papers seem to support good effectiveness and tolerability of anti-CGRP-mAbs in the real-world setting. There is an unmet need for large prospective real-world studies providing long-term follow-ups of patients treated with anti-CGRP-mAbs.

## 1. Introduction

For decades, the pharmacological prophylaxis of migraines has been based on medications that were non-specific for migraines, which led to low adherence rates due to limited efficacy and poor tolerability [1]. Monoclonal antibodies against the calcitonin gene-related peptide (CGRP) or its receptor (anti-CGRP-mAbs) have opened a new era for migraine prevention.

CGRP is a neuropeptide also acting as neurotransmitter that has, among others, a crucial role within the pathophysiology of migraines. Its release is increased during migraine attacks [2] and intravenous infusion of CGRP can trigger migraine-like attacks in migraine patients. CGRP is a very potent vasodilator and exerts its action not exclusively in the brain. It contributes to reactive vasodilation during myocardial infarction and vasospasms during subarachnoid hemorrhages. It is involved in the transmission of pain and sensory stimuli, in wound healing, and it has functions in the gastrointestinal system [3]. 

Phase 2 and phase 3 trials showed no signs of an increased incidence of vascular events or vascular complications in patients under therapy with an anti-CGRP-mAb. Moreover, package information leaflets do not list any vascular disease or risk factor as contraindications against these antibodies. Nonetheless, these leaflets contain warnings to be cautious in patients with a history of cardiovascular or cerebrovascular diseases.

Anti-CGRP-mAbs are effective in episodic [4,5,6,7,8] and chronic migraines [9,10,11,12], including difficult-to-treat patient groups with multiple treatment failures, psychiatric comorbidities [13,14,15,16,17,18], or medication overuse [19,20,21,22]. Outcome measures involve monthly days with migraines, any headache and use of acute medication, the 50% responder rate (i.e., the proportion of patients experiencing a reduction in monthly migraine days by 50% or more), as well as functional and patient-related outcomes [23,24,25,26]. 

The CGRP-antibodies fremanezumab and galcanezumab as well as the CGRP-receptor antibody erenumab, all of which are administered subcutaneously, have been licensed for migraine prevention since 2018. More recently, eptinezumab was licensed, another CGRP-antibody, which is administered intravenously. Instead of a daily intake of medication, as required for standard pharmacoprophylaxis, anti-CGRP-mAbs are administered once every four weeks, every month, or every three months. 

Altogether, they are approved for episodic migraines with at least four migraine days per month, and chronic migraine. Reimbursement regulations differ from country to country. This leads to different uses in daily clinical practice, with respect to the number of previously prescribed prophylactic medications, necessity of therapy breaks, or switches from one antibody to another.

While some long-term studies, mostly open-label extensions of phase 2 or phase 3 studies in highly selected populations, are reassuring concerning safety [27,28,29,30], real-world evidence in unselected patient groups is of particular interest. Issues deserving further study in the real-world setting include long-term safety and effectiveness, impact on migraine auras, outcomes of pausing the treatment and of switching to another antibody, and data in special groups (such as elderly persons and patients with comorbidities).

Since the approval of anti-CGRP-mAbs, plenty of studies and case reports dealing with real-world experience and focusing on various aspects of these antibodies have been published. The aim of this article was to gather real-world data on anti-CGRP-mAbs and to review these data systematically with respect to pharmaco-epidemiological findings, headache diagnoses, general effectiveness, effectiveness in patients with previous treatment failures, differences in effectiveness of the antibodies, outcomes of pausing treatment, switching to another antibody, and discontinuing treatment, as well as tolerability and safety.

## 2. Methods

### 2.1. Search Methods

We performed a review of the literature using PubMed, concerning real-world studies of migraine patients treated with anti-CGRP-mAbs. Search terms included the following: erenumab, fremanezumab, galcanezumab, eptinezumab, CGRP, calcitonin, real, case, migraine, vertigo, cyclic vomiting, and visual snow. To focus the results, we conducted 8 individualized searches: 2 for each monoclonal antibody—one using the keyword real and one search using the keyword case.

### 2.2. Selection Criteria

Our selection criteria were language (English), primary headache type (migraine and migraine-related disorders), and study design (real-world data). The last search took place on 1 December 2022.

### 2.3. Review Preparation and Statistics

The systematic review was prepared according to the latest PRISMA (Preferred Reporting Items for Systematic Reviews and Meta-Analyses) guidelines [31], and study data were gathered into an Excel table. Descriptive statistics were conducted in IBM SPSS Statistics 21.

## 3. Study Characteristics

Our search yielded 251 results from the eight individual searches. After we applied selection criteria and excluded duplicates, 145 articles remained for hand-search to exclude additional nonrelevant publications. Finally, we included 134 articles in this review. An exact breakdown of the search results can be seen in Figure 1.

We classified these articles into pharmacoepidemiologic studies (n = 8) [32,33,34,35,36,37,38,39], clinic-based studies (n = 83) [40,41,42,43,44,45,46,47,48,49,50,51,52,53,54,55,56,57,58,59,60,61,62,63,64,65,66,67,68,69,70,71,72,73,74,75,76,77,78,79,80,81,82,83,84,85,86,87,88,89,90,91,92,93,94,95,96,97,98,99,100,101,102,103,104,105,106,107,108,109,110,111,112,113,114,115,116,117,118,119,120,121,122], case reports (n = 40) [123,124,125,126,127,128,129,130,131,132,133,134,135,136,137,138,139,140,141,142,143,144,145,146,147,148,149,150,151,152,153,154,155,156,157,158,159,160,161,162], and other articles (n = 5) [163,164,165,166,167]. Eighty-nine articles were retrospective [32,33,34,35,36,37,38,39,40,41,42,43,44,45,46,47,48,49,50,51,52,53,54,55,56,57,58,59,60,61,62,63,64,65,66,67,68,69,70,71,106,109,111,117,118,120,123,124,125,126,127,128,129,130,131,132,133,134,135,136,137,138,139,140,141,142,143,144,145,146,147,148,149,150,151,152,153,154,155,156,157,158,159,160,161,162,163,164,165,166,167] and 45 prospective [72,73,74,75,76,77,78,79,80,81,82,83,84,85,86,87,88,89,90,91,92,93,94,95,96,97,98,99,100,101,102,103,104,105,107,108,110,112,113,114,115,116,119,121,122]. Outcomes for erenumab, galcanezumab, and fremanezumab were reported in 113, 45, and 31 studies. Real-world data of eptinezumab were only available in one study [167].

## 4. Pharmacoepidemiologic Studies

Table 1 summarizes the pharmacoepidemiologic studies that looked at real-world prescription data. Due to the nature of such databases, clinical outcomes such as efficacy, adverse events, or days with acute medication use could not be collected. However, large insurance-based datasets allowed us to look at physicians’ prescription patterns or claims made by the patients. Thus, the persistence of treatment and adherence could be assessed. Inferences on the efficacy of the therapies could only be made indirectly. 

We grouped the main study results by outcome parameters and looked at prescriptions of acute and prophylactic migraine medications, treatment adherence, health care resource utilization (HCRU), days with sick-leave, and impact of migraine and adverse events.

### 4.1. Acute Medication

Five studies assessed the prescription of acute migraine medications six to twelve months before and six to twelve months the after first administration of an anti-CGRP-mAb [32,33,35,37,38]. The different methods of data representations do not allow us to calculate direct comparisons or summaries of data. Comparing baseline to treatment with erenumab, the prescription of acute migraine medications decreased by 49% [35] and by 23% [38], respectively; and the proportion of patients using no prescription acute medication at all or only one type increased [33]. Analyzing specific acute migraine medications, the prescription of non-steroidal anti-inflammatory drugs decreased significantly [37]. In addition, there was a (numerical) decrease in the prescription of triptans [32,37] and barbiturate-containing acute medications [37]. Comparing erenumab to OnabotulinumtoxinA, reductions were stronger for erenumab [37].

### 4.2. Prophylactic Medication Apart from Anti-CGRP-mAbs

Four studies assessed the prescription of prophylactic medications before and after the first administration of an anti-CGRP-mAb. Three studies reported on erenumab [32,33,35] and one included erenumab, fremanezumab, and galcanezumab [39]. In the first, the prescription of other prophylactics decreased by roughly 30%. In addition, this study found that 50% percent of the patients with standard therapies stopped them within one month, but less than 20% of the patients on anti-CGRP-mAbs stopped their antibody-therapy within one month [32]. The second study [33] observed a shift to fewer prescriptions of preventive medications. The mean time until other ongoing preventive medications were stopped was 185 to 230 days, and 36% had stopped other prophylactics at twelve months in the third study [35]. In the study including three antibodies [39], patients received significantly less often other prophylactics during follow-up and 75% stopped other prophylactics during the twelve-month follow-up.

### 4.3. Adherence and Persistence

Three studies examined the adherence or persistence. [34,35,39] The adherence to anti-CGRP-mAbs was higher (≥0.8) than to oral prophylactics but still not at the optimum [35]. In the Novartis Go Program [34] offering advice, injection training, and erenumab free of charge until the individual insurance was willing/able to pay for erenumab, the persistence of treatment was 71% at 360 days and 63% at 450 days, which is better than under oral preventives [1]. Varnado et al. [39] found a higher main persistence under anti-CGRP-mAbs than under standard prophylactics and a significantly higher adherence at six and twelve months (medication possession rate 58% vs. 37%, proportion of days covered 55% vs. 35%).

### 4.4. Health Care Resource Utilization

HCRU was analyzed in four studies [32,36,37,38]. During treatment with erenumab, migraine-specific office visits decreased statistically significantly from 86.2% to 77.6% [38], claims for health care utilization decreased by 10–19% [37], and health care visits decreased by 45% in the study of Autio et al. [32]. Similarly, treatment with [36] was associated with a significant reduction in HCRU. Emergency visits decreased by 25% and outpatient visits by 22%.

### 4.5. Sick-Leave and Impact of Migraine

Only one very small study [32] addressed the question of whether treatment with an anti-CGRP-mAb, namely erenumab, has an impact on sick-leave. The results suggest that erenumab may significantly reduce the number of headache-related sick-leave days in employed patients with migraines, managed in routine clinical practice. In detail, sick leave days per patient year decreased by 74%, i.e., from 4.9 to 1.3. 

Another single study [36] found a reduction in self-reported headache frequency and migraine pain intensity during treatment with fremanezumab.

### 4.6. Summary

These pharmacoepidemiologic data indirectly hint to the real-world effectiveness of and adherence to anti-CGRP-mAbs. The biggest limitation is that clinical outcome data were not available. Most of these studies were carried out in the Unites States of America or Canada, only one in Europe, reflecting the insurance systems of these countries which cannot be generalized to other countries. The observation periods were limited to 6 to 12 months. 

Such databases capture the prescription of medications and the dispensation to patients; however, they cannot capture if the medications are actually used by the patients, and they were not primarily made for research. Moreover, pharmacoepidemiologic data do not provide information on the reasons for stopping therapy with an anti-CGRP-mAb. 

All but four pharmacoepidemiologic studies included only erenumab, which was marketed first [32,33,34,35,37,38]. All studies bear the risk of bias, as they were supported by pharmaceutical companies. The risk of bias is highest in the studies by Varnado et al. [39] and Gladstone et al [34]. The first, reporting claims data of erenumab, galcanezumab, and fremanezumab [39], was performed by Eli Lilly and focused on the switch to galcanezumab. The second [34] was biased because Novartis offered erenumab for free if the patient’s insurance did not cover the costs.

## 5. Clinic-Based Studies

As of 1 December 2022, we found 83 clinically based, real-world studies involving all anti-CGRP-mAbs except for eptinezumab. Details of all studies are given in Appendix A. Out of the 83 studies, 21 were supported by pharmaceutical companies.

### 5.1. Study Design

About half of the studies had a prospective study design (45/83), stating more often clear inclusion (77/83) and to a lesser degree, clear exclusion criteria (41/83). All but one study cited the latest ICHD-3 criteria [168], while about half of the studies stated whether the migraine patients had auras or not—26 did not make this distinction. Practically all studies recruited patients with chronic migraine (81/83) and just over half of these included patients with episodic migraine (49/83—no study focused solely on episodic migraine). Medication overuse headache (MOH) was clearly reported in sixty of these, three did not, while twenty studies did not explicitly state the presence of MOH patients.

### 5.2. Patients

On average, 180 patients (SD 269.6, median 100, IQR 52–160) were recruited. As can be expected, most of these patients were women (mean 149 patients, SD 220, median 85, IQR 41–132); however, four studies did not specify gender. The average age of the patients was 46.7 (median 47.1, IQR 45.7–49), although some studies opted to report median and IQR instead. All but four studies reported patients as having prior prophylactic treatment failure or refractory migraines. Unfortunately, many of these studies lost their patients during the study period. In 73 articles reporting patient numbers at baseline as well as at the last available follow-up, the total number of patients decreased by a mean of 18.9%. Driessen et al., in both of their papers [49,111], went on to lose over 90% of the initially recruited patients (1003 recruited and 92 patients analyzed at 6 months of treatment). Thus, the reported results must be considered critically.

### 5.3. Anti-CGRP-mAbs

Erenumab alone was studied in 48/83 articles, eight studied galcanezumab alone, three examined exclusively fremanezumab; meanwhile seven studies compared the effects of erenumab and galcanezumab, three studied examined patients treated with erenumab and fremanezumab, and 14/83 studies included all three.

### 5.4. Effectiveness

The effectiveness of anti-CGRP-mAb treatment was reported in 77 of the 83 clinic-based studies; however, the data is grossly heterogeneous. Only 16 studies reported both monthly migraine and monthly headache days. Baseline average monthly migraine days were not reported by 52 studies; instead, 13/52 studies opted to report median and IQR. The other 39 decided to split their results in terms of either anti-CGRP responders or non-responders, episodic migraine or chronic migraine, or did not report this data at all. The average number of monthly migraine days at 3 months of treatment with an anti-CGRP-mAb was reported by just thirteen studies, at 6 months by eleven studies, none reported at 9 months of treatment, while four reported average monthly migraine days after 12 months of treatment. Similarly, the average number of monthly headache days was also inconsistently reported. Only 29 studies reported baseline monthly headache days and this number dwindled with the respective 3-month, 6-month, 9-month, and 12-month follow-ups (thirteen, ten, one, and four studies, respectively). Another effectiveness metric, monthly acute medication use, was comparably inconsistently reported. Only twenty studies reported baseline data, which went on to be sparsely reported, with only five studies reporting 12-month data. Finally, 50% responder rates (≥50% reduction in monthly migraine/headache days compared to baseline) were reported in 71/83 articles—however, again with varying methodologic preference. The 50% responder rates in terms of monthly migraine days at specific time points, namely 3, 6, and 12 months were reported only in twenty-two, thirteen, and eight articles, respectively. The average proportion of 50% responders seemed to increase over time and was 44% (SD 20.1%, median 48.7%, IQR 27.5%–58.3%) at 3 months, 49.7% (SD 27.1%, median 53.3%, IQR 26.8%–67.1%) at 6 months, and 63.6% (SD 25.6%, median 61.1%, IQR 46.6%–91.1%) at 12 months. A similar trend could be seen in the 50% responder rate in terms of monthly headache days. A summary of the effectiveness of the different anti-CGRP-mAbs is provided in the Appendix A.

The overall conclusion is that a significant treatment benefit is reported in the real-world longitudinal studies, just as in the Phase 3 open-label extensions [27,28,29,30]; however, these real-world results must be treated critically as many studies are limited by their short observation period and many lost patients to follow-up, which significantly affected the responder rates reported; i.e., non-responders are probably more likely to be lost during follow-up than responders, and thus the response rate will increase. Moreover, 34 of the 56 studies did not include baseline data and therefore, it was impossible to verify the authors’ claimed observed effectiveness rates.

### 5.5. Concomitant Pharmacoprophylaxis

Around half of the studies (42/83) also tracked whether patients remained on previous migraine prophylaxis while undergoing treatment with an anti-CGRP-mAb. Thirteen studies conducted direct comparisons to treatment with OnabotulinumtoxinA, after which antidepressants were the next most common concomitant prophylactic reported. Patients treated concomitantly with OnabotulinumtoxinA showed significant reductions in migraine and headache days, displaying a possible synergistic benefit of the two treatments in patients with chronic migraine. None of the 42 articles clearly stated whether the concomitant prophylactic treatment was slowly titrated out or whether they were regular migraine therapies. Thus, the real-world data do not allow us to infer whether concomitant prophylactic migraine treatment works synergistically to relieve the burden of disease in migraine patients.

### 5.6. Treatment Break

Two studies described patients undergoing planned and unplanned treatment breaks [47,68], ten explicitly described a planned break in treatment with the anti-CGRP-mAb [52,62,73,80,83,84,92,95,101,122], and six reported an unplanned break in treatment [42,45,58,66,74,96]. In contrast, 65 of these 83 real-world studies did not have study periods that allowed for analysis of a treatment break or did not describe a treatment break at all. Most interestingly, all but one of the studies addressing planned treatment breaks made their primary endpoints the effect of pausation of treatment, which meant little was discussed about their treatment benefit leading up to the treatment break [52,62,73,80,83,84,92,95,101]. Nine studies reported the time to migraine return and the corresponding patient number [47,52,74,80,83,84,92,101,122]. Eight found that in a range from 4 to 12 weeks after pausing or interrupting treatment with anti-CGRP-mAbs, patients began to experience increased migraine frequency [47,52,74,80,83,84,92,122]. Vernieri et al. reported no worsening of migraine frequency within the first 3 months [101]. In this regard, the studies by Gantenbein et al. [52], Iannone et al. [85], and Nsaka et al. [122] give us the most relevant real-world data, as they shared the initial 12-month treatment benefit in addition to the effects of treatment pausation of 3 months and 1 month after re-initiation. Gantenbein et al. and Nsaka et al. reported that no participants experienced lasting effects (i.e., longer than 3 months) of their anti-CGRP therapy [52,122], while Iannone et al. reported that 12/44 patients did not meet criteria to restart anti-CGRP therapy [84].

### 5.7. Switching to Another Anti-CGRP-mAb

Of the 19 studies looking at ≥2 anti-CGRP-mAbs, 11 studies considered the effects of switching therapies. These studies examined a variety of questions without consistent reporting. The overarching aims were to reaffirm effectiveness and safety of the studied anti-CGRP-mAbs and to compare them against other prophylactic treatments (i.e., OnabotulinumtoxinA). In general, the clinical aspects of anti-CGRP-mAb treatment appear very heterogeneous. Two studies documented an improvement after switching to another anti-CGRP-mAb; 8/25 [62] and 8/15 [65] patients showed a ≥30% improvement in monthly migraine days after switching from anti-receptor-mAb to an anti-ligand-mAb.

### 5.8. Discontinuation of Antibody Treatment

Many studies discussed treatment discontinuation (57/83). Interestingly, 26 studies had no patients discontinue treatment. An often-cited reason for discontinuation was “perceived lack of effectiveness”; however, no paper went on to state the migraine or headache frequencies of these patients.

### 5.9. Adverse Events

Sixty-one studies reported adverse events, eighteen saw no adverse event in their patient populations, and four did not give any information on adverse events (Table 2). Studies mainly relied on patient reporting of adverse events (61/83), while one went further and used a structured questionnaire. Adverse event intensity and duration were rarely gathered (five and eight articles, respectively). Causality of the adverse event with anti-CGRP treatment was discussed in 55 articles and adverse event frequency (i.e., how many patients) was mentioned in 57/83 articles. Constipation was the most common side effect reported, while reaction at the site of injection was the next most common. A plethora of other adverse events was reported in the studies that are not part of the official list of side effects for anti-CGRP-mAbs. Among these, flu-like symptoms, arthralgia, gastric pain, and chest pain were more frequent. In addition, there were single observations of hypertension and hair loss. Forty-four of the eighty-three articles described the cessation of treatment due to adverse events. Generally, an average of 5.9% of the patients (SD 11.4%, median 1.2%, IQR: 0–5.9%) stopped treatment due to side effects.

Recently, a prospective study from the Leiden Headache Center reported a small blood pressure increase in migraine patients after initiation of erenumab or fremanezumab [116]. In this study, the effect was more consistent after erenumab initiation, where systolic blood pressure was elevated in all follow-up visits, whereas only systolic blood pressure was elevated in the first follow-up visit on fremanezumab. No blood pressure increase was observed in a control group without CGRP treatment. However, this study contrasts with pivotal erenumab and fremanezumab phase-3 studies and open-label extension studies [4,9,11,13,14,15]. No blood pressure increase was observed in an open-label study over 5 years [27]. Methodological issues such as the standardization of blood pressure measurements and the balancing of investigational groups merit discussion. In summary, a subtle signal for the development of worsening of blood pressure after CGRP blockade is possible in the real-world setting, but further investigation is needed. Thus, repeated blood pressure measurements can be recommended for patients on anti-CGRP-mAb therapy.

### 5.10. Severe Adverse Events

We found 18 articles that reported one or multiple severe adverse events, 39 that found none, and 26 that did not make any mention of severe adverse events. The most common severe adverse event reported was severe constipation; no deaths were directly attributed to the therapy.

### 5.11. Summary

The results from clinically based, real-world studies are diverse and generally did not have reporting guidelines to refer to until recently [169,170,171]. This lack of reporting guidelines—or at least lack of awareness—has led to a variety of data to be published since the approval of anti-CGRP-mAbs. Nevertheless, clinic-based real-world studies seem to suggest that the monoclonal antibodies are similarly effective as seen in the clinical trials. Furthermore, their safety and tolerability profiles appear to be equally similar; except, for hypertension being added to the official list of possible side effects, even though the causal relation is disputed [163,164].

## 6. Case Reports

Among forty case reports, twenty-seven described a single patient, four reported on two, and five on three patients, and one paper each included four, five, eight, and ten patients [123,124,125,126,127,128,129,130,131,132,133,134,135,136,137,138,139,140,141,142,143,144,145,146,147,148,149,150,151,152,153,154,155,156,157,158,159,160,161,162]. In these 77 patients, the mean age was 44.6 (SD 9.63) and 76.6% were women, 30 had used erenumab, 12 had used fremanezumab, and 6 had used galcanezumab. 

Case reports may give hints on rare adverse events in the clinical setting. Inherently, causal associations between single observations and a given drug can hardly be drawn, but collecting information is important to detect the possible clustering of events. Notably, beneficial effects of anti-CGRP-mAbs beyond their actual indication are also possible. The fact that most reports were on erenumab, the first anti-CGRP-mAb to be licensed, may give a biased view on effects or side effects. Furthermore, most reports were on observations in women, reflecting prescription practice and migraine epidemiology. Conceptually, case reports were found to cover the following situations:i.Improvement of a symptom or comorbid condition;ii.Effectiveness and no adverse events under special circumstances;iii.Adverse events in otherwise healthy individuals;iv.Adverse events because of possible drug interactions, or potentiation of side effects;v.Deterioration of preexisting disorder.

Improvement of a symptom or comorbid condition with anti-CGRP-mAbs was reported for migraine aura [124], cluster headache [134,148], headache related to sexual activity [139], nummular headache [160], restless leg syndrome [159], sleep terrors [149], and stuttering [150]. In three patients, severe nausea induced by erenumab led to smoking cessation [140].

Single reports on effectiveness without adverse events covered the exposure to erenumab in the first weeks of pregnancy [129,157], during whole pregnancy [155], during breast feeding [133], and in myasthenia gravis treated with immunoglobulins [143]. A case-report of three pregnancies reported two full-term deliveries and one miscarriage after exposure to erenumab. The two full-term pregnancies administered one dose of erenumab during the pregnancy, immediately stopping treatment afterwards and not experiencing any complications, while the patient who had a miscarriage ceased treatment 1 month prior to learning she was pregnant. In the latter case, a rare intrauterine complication was found (gestational throphoplastic neoplasia), but the known risk factors for this complication do not seem to correlate with the mechanisms of erenumab [155]. According to a WHO pharmacovigilance database on erenumab, galcanezumab, and fremanezumab exposure during pregnancy and lactation, no specific risk for toxicity could be detected, but data were limited to 94 cases (more than half on erenumab). Although adverse events including spontaneous abortions or birth defects were reported, this was not increased in the exposed patients. Further data collection, for instance, in registries, seems mandatory before definite advice concerning the safety of CGRP-mAbs in pregnancy and lactation can be given [172].

In one patient each, erenumab and fremanezumab were effective in COVID-19-related migraine exacerbations [126,132], and in two patients, the use of rimegepant during treatment with erenumab was found effective and was well-tolerated [142]. Notably, no recommendation concerning the safety in these conditions can be given based on this anecdotal evidence. In contrast, a series of 10 patients treated with both erenumab and OnabotulinumtoxinA added to the pharmacoepidemiologic data on this combination [147]. In the absence of evidence from RCTs, patients with otherwise refractory migraine, may benefit form anti-CGRP-mAbs administered together with OnabotulinumtoxinA.

A possible anti-CGRP-mAb adverse event in an otherwise healthy individual was reported by Rozen et al. [145]. In summary, a 43-year-old woman developed a sexual headache and a thunderclap headache 2 days after the second dose of erenumab and after high-altitude exposure and triptan use in the week before. CT angiogram results showed narrowing of the left middle and anterior cerebral arteries, consistent with reversible cerebral vasoconstriction syndrome. Treatment with erenumab and triptans was stopped, and verapamil was initiated. The CT angiogram was normal 4 weeks after initial neuroimaging, supporting the diagnosis of reversible cerebral vasoconstriction syndrome (RCVS). [145]. The observation of cerebral vasospasms after a CGRP blockade is of considerable interest, given the vasodilatory effects of CGRP. However, it has been suggested that anti-CGRP-mAbs might not reach the abluminal compartment of cerebral blood vessels within the blood brain barrier and thus might be an unlikely cause of RCVS [173,174]. Another limitation of the hypothesis of a possible causal relationship between erenumab and RCVS is pharmacokinetics, as the maximum concentration of erenumab is reached later. From a clinical point of view, this case report contrasts with the patient mentioned above who used erenumab for migraines and experienced improvement of headaches related to sexual activity. 

An Australian-Irish collaboration found serious adverse events (SAEs) in eight patients from centers in Australia and Ireland, forcing all patients to cease their use of anti-CGRP-mAbs, related to inflammatory complications of CGRP monoclonal antibodies [161]. In this article, three of the eight patients had a pre-existing, well-controlled rheumatological or dermatological disease, which worsened significantly in eight patients after the anti-CGRP-mAb therapy was started. Six patients developed a de novo inflammatory disease after exposure. Causality was established based on the remission of symptoms after withdrawal of anti-CGRP-mAbs. Patient 1, for example, suffered from rheumatoid arthritis, dyslipidemia, and pulmonary fibrosis, finally experiencing autoimmune hepatitis after one injection of erenumab and ceasing therapy thereafter. One patient with fibromyalgia and chronic fatigue syndrome developed ocular Susac’s syndrome after 12 months of erenumab treatment. In this series, one patient without significant comorbidities developed granulomatosis with polyangiitis after treatment with fremanezumab. Patients 6 and 7, on the other hand, experienced worsening of their psoriatic conditions, leading them to stop their therapies with galcanezumab and erenumab, respectively. The cases provided and explained by Ray and colleagues show that antagonism of CGRP should be carefully considered, especially in patients with pre-existing immunological diseases, as CGRP’s role in inflammatory regulation should not be underestimated, and its inhibition can lead to serious, albeit rare, SAEs. The authors discussed possible effects of CGRP blockades on Langerhans cells, macrophages, and mast cells, as well as effects on cytokine production. In this real-life series with SAEs, causality could not be further undermined since re-exposure was not possible; thus, it is vital that such events be consistently reported. While these case series are extremely engaging, reports regarding the real-life, complication-free use of anti-CGRP-mAbs in patients with pre-existing autoimmune conditions should be reported—and are equally valuable.

A further interesting article by Wurthmann et al. reported skin lesions and impaired wounds in a previously healthy patient [151]. In essence, the patient using erenumab presented with crescent-shaped necroses on the inner surface of the left forearm that formed from a singular erythematous papular skin lesion, no bigger than 1 cm. The vessels supplying the upper cervicobrachial plexus were thrombosed, and the authors hypothesized that erenumab caused a decreased blood flow to small blood vessels, leading to necrosis. Whether remission of the symptoms following cessation of erenumab supports this hypothesis must remain open. 

In a case report by Aradi et al. [125], it is less clear if the patient was otherwise healthy. The authors describe a 41-year-old woman with migraine without aura who developed a right thalamic infarction following a first dose of erenumab. The stroke developed 34 days after the first exposure to erenumab and 4 h after medication with rizatriptan, which the patient had taken before without complications. In addition, the patient was on a low-dose estrogen oral contraceptive. She had no other vascular risk factors. A CT angiography of the head and neck demonstrated a proximal right posterior cerebral artery stenosis in the P1 segment, which resolved after 2 months and was thus interpreted as a vasospasm. In this patient, blood tests for hypercoagulopathy were negative and transesophageal echocardiography revealed no source of embolus; however, long-term electrocardiograms to rule out atrial fibrillation were not reported. Thus, this case is potentially confounded by incomplete diagnostic work up and concomitant use of other substances potentially related to ischemic stroke. The authors discussed the possibility that CGRP blockades might impair vasodilatory mechanisms to compensate for triptan-induced vasoconstriction. However, triptans seem to reverse vasodilatation of intracranial arteries during the migraine attack rather than cause intracranial vasoconstriction [175] and have been safely used for migraine therapy for decades.

The case report from Lehman et al. describing deterioration of a pre-existing cerebrovascular disorders warrants serious scrutiny [138]. An anti-CGRP-mAb was prescribed to a migraine patient with cerebral proliferative angiopathy. Two days after the first subcutaneous administration of erenumab, the patient presented with status epilepticus and showed diffusion abnormalities in the MRI in vicinity to the cerebral proliferative angiopathy. The authors summarized that the patient had recurrent refractory epilepsy with lasting damage to his motor as well as visuospatial functions. This report serves to teach that anti-CGRP-mAbs should be prescribed with caution, weighing the risks and benefits of anti-CGRP-mAbs in certain comorbid conditions. 

More case reports on adverse events are summarized in Table 3.

### Summary

Based on anecdotal evidence from case reports, no definite conclusions can be drawn. 

Case reports included observations of contradictory findings, e.g., de novo appearance [136] or substantial improvement of auras [124] and de novo appearance or significant improvement [139] of headaches related to sexual activity [145]. This could be explained by differential effects based on unknown cofactors or reflect the report of mere coincidences. Based on current real-world data, no clustering of rare side effects was observed. 

However, in our opinion, the observation of possible adverse events related to the blockade of the vasodilator CGRP deserves attention. One stroke related to vasoconstriction [163] and one case of RCVS [145] were reported. It must be emphasized that cryptogenic stroke is common in young individuals. In addition, new appearance or exacerbation of Raynaud´s phenomenon was observed [128]. The issue of the possible development or exacerbation of hypertension is not fully understood yet. Thus, we conclude that patients should be screened for high vascular risk before the initiation of CGRP-based therapies.

## 7. Other Articles

Finally, we want to review five articles: two related to hypertension [163,164], two articles focusing on adverse events (AEs) [165,166], and one reporting on Raynaud’s phenomenon [167] in patients using anti-CGRP-mAbs. 

Saely et al. summarized 57 reports of elevated blood pressure associated with the use of erenumab submitted to the FDA Adverse Event Reporting System [164]. In this case series, baseline blood pressure was reported in only half the patients, and reports of hypertension were based on single elevated blood pressure measurements, which precludes robust conclusions. Subsequently, Dodick et al. gathered information on all post-marketing adverse event reports of hypertension in erenumab users using the Amgen global safety database and summarized them into a single article containing 355 patient cases [163]. Adverse events of hypertension occurred, in part, in patients with pre-existing hypertension—one third of patients with serious hypertension had previous hypertension. Time of onset was not described in more than half of the reports, while about half of the cases with hypertension were reported after 1 week of the first administration. The authors conclude that adverse event rates of hypertension reported with erenumab in the post-marketing setting were generally low and that only with additional studies can this risk be properly characterized.

Two studies focused specifically on adverse events during real-world use of anti-CGRP-mAbs [165,166]. Overall, patients reported “migraine”, “headache”, and “drug ineffective”, along with migraine-associated symptoms (i.e., nausea) and “injection-site” reactions as the most common AEs for all erenumab, galcanezumab, and fremanezumab. Cardiovascular events were outside of the top ten AEs for any of the three anti-CGRP-mAbs. “Constipation” was the second most commonly reported AE for erenumab; however, it found itself outside the top ten AEs for fremanezumab or galcanezumab. Serious AEs were infrequent across all three anti-CGRP-mAbs [166]. A particular topic of interest was Raynaud’s phenomenon (RP), which is followed by the World Health Organization in its VigiBase® [165]. CGRP-targeting drugs were significantly associated with Raynaud’s phenomenon. Erenumab was the most reported anti-CGRP-mAb (with 56/99 reports). The median time to RP onset was 84 days; however, it never led to fatality, with one patient suffering gangrene and extremity necrosis [165]. The authors could not, however, conclusively determine from the evidence in the database whether the occurrence of RP was de novo or a worsening of pre-existing RP. Nevertheless, consideration should be taken before prescribing anti-CGRP-mAbs to migraine patients with the potential to develop RP.

Breen et al. [167] examined a cohort of patients with Raynaud’s phenomenon from a specialized clinic who were treated with CGRP antagonists for migraines. Most Raynaud patients (160/169) experienced no complications, and a minority (9/160) of patients experienced complications including microvascular complications (such as worsening facial telangiectasias or digital necrosis requiring surgery), all of whom had received anti-CGRP-mAbs (erenumab, galcanezumab, fremanezumab, and eptinezumab). Approximately half of patients with complications developed Raynaud’s phenomenon de novo shortly after the first exposure. In this cohort, no significant difference in demographic or clinical variables was detected in patients with or without complications. The authors concluded that anti-CGRP-mAbs should be used with caution in patients with Raynaud’s phenomenon.

## 8. Conclusions

With few exceptions, available real-world data are limited by retrospective data collection, small patient numbers, and short follow-up periods. For the time being, the majority of real-world papers seem to support good efficacy and tolerability of anti-CGRP-mAbs in the real-world setting. Furthermore, direct head-to-head comparisons between the anti-CGRP-mAbs are made difficult by the heterogeneity of results reported. Reports of rare adverse events must be carefully monitored, but causal relations may not be concluded from single case studies. Particular attention is given to vascular events related to anti-CGRP-mAbs, although no clear vascular safety signal has emerged yet. De novo appearance or worsening of Raynaud’s phenomenon must be carefully monitored. There is an unmet need for large prospective real-world studies and registries providing long-term follow-ups of patients treated with anti-CGRP-mAbs.

## Figures and Tables

**Figure 1 cells-12-00143-f001:**
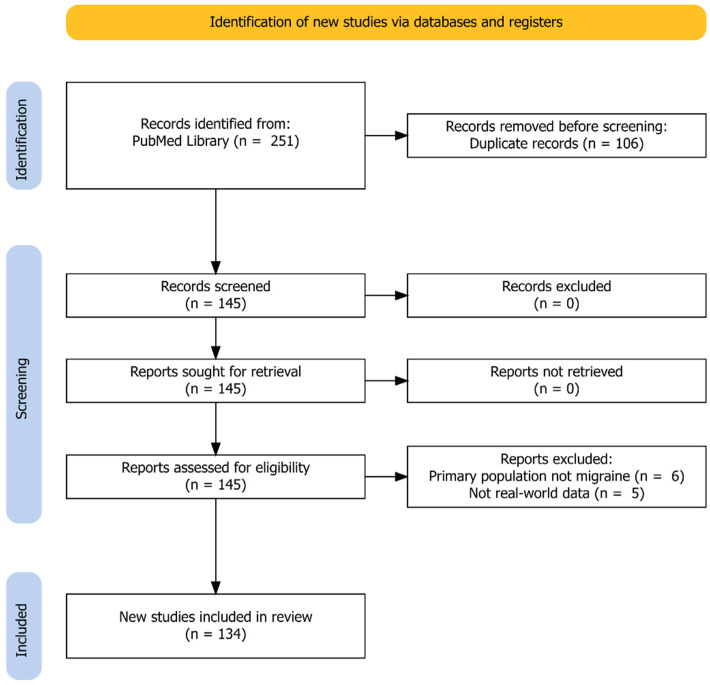
Identification of studies according to the PRISMA Guidelines (Preferred Reporting Items for Systematic Reviews and Meta-Analyses).

**Table 1 cells-12-00143-t001:** Pharmacoepidemiologic studies.

Reference	CGRP-mAb	Patients (n)	Women (%)	Mean Age (Years)	Migraine Diagnosis Available/Diagnosis According to	Inclusion of Patients with				
						Migraine with Aura	Chronic Migraine	Medication Overuse	Prior Treatment Failure	Other Prophylactic Medication
[32]	E	82	85.4	45	Yes/ICD-10	NA	NA	NA	Yes	Yes
[33]	E	4437	85.8	47	Yes/ICD-10	Yes	Yes	NA	NA	Yes
[34]	E	14,282	83.0	46	No	Yes	Yes	NA	Yes	NA
[35]	E	29,451	79.2	47	No	NA	NA	NA	Yes	Yes
[36]	F	172	83.7	46	No	NA	NA	NA	Yes	Yes
[37]	E, OBTA	2676	91.6	50	Yes/ICD	Yes	Yes	NA	Yes	Yes
[38]	E	3171	84.8	51	Yes/ICD	Yes	Yes	NA	Yes	Yes
[39]	E, F, G	3082	85.7	44	Yes/ICD-10	Yes	Yes	NA	Yes	Yes

Abbreviations: E erenumab, F fremanezumab, G galcanezumab, OBTA OnabotulinumtoxinA, ICD International Classification of Diseases, ICD-10 International Classification of Diseases, Tenth Revision, NA: information not available.

**Table 2 cells-12-00143-t002:** Most frequent adverse events reported in clinic-based studies.

Adverse Event	Inquired (Number of Studies)	Observed (Number of Patients)
Constipation	50	1251
Reaction at injection site	42	217
Dizziness	39	78
Muscle cramps	38	41
Pruritus	37	44
Pain at injection site	36	76
Skin rash	36	19
Urticaria	36	12

**Table 3 cells-12-00143-t003:** Adverse events from case reports.

Adverse Events in Otherwise Healthy Individuals					
Ref.	Age	Sex	Exposure	Adverse Event	Comment
[123]	54	M	G	Erectile dysfunction	More than 2 months after start, reversible after 2 half-lives, rare use of metoprolol for palpitations
[128]	33	M	E	Raynaud’s phenomenon	When in the cold cca. 1 h, had RP of all the fingers and toes bilateral with temperature change and numbness lasting about 1 h
[131]	38	F	E	Restless leg-like symptoms	De novo symptoms; erenumab continued despite symptoms
[131]	47	F	G	Restless leg-like symptoms	De novo symptoms; cessation of symptoms after erenumab discontinuation
[136]	61	F	G	Migraine aura	Unsuccessful with erenumab, 1 month after last injection switch to galcanezumab (240 mg loading dose, followed by a maintenance dose of 120 mg 28 days later), within 1 week after the first dose of 120 mg, experienced first visual aura
[137]	48	F	G	Skin lesions in fixed location	After several months, developed erythema and pruritus of left upper arm within 24 h of self-injection (lasting up to 3 days), evolved into a nonpruritic, non-painful, chronic, brown-to-blue patch. Each monthly injection of galcanezumab resulted in same clinical course (at identical site on the left arm), despite injecting different areas on body (incl. the abdomen and thighs), without reaction at injection site
[141]	52	F	F	Non-immediate rash	Causal relation confirmed with pinprick test
[144]	26	F	E	Stypsis	Exteroceptive suppression period of the temporalis muscle was assessed during a ten-day washout period, before starting erenumab and after 4 months of erenumab treatment
[146]	60	F	E	Xerostomia	After first injection, reported dry mouth in the next ten days; similar duration after 2nd injection
[151]	51	F	E	Impaired wound healing of trivial injury	Improvement after discontinuation of erenumab
[156]	57	F	E	Myocardial infarction	Former smoker, family history of cardiovascular disease
[162]	55	M	E	Myocardial infarction	BMI of 29, non-smoker, suffered from hypertension, dyslipidemia, and prior myocardial infarction in 2012
[154]	48	F	E	Symmetrical drug-related intertriginous and flexural exanthema	Erenumab discontinued and switched to fremanezumab
**Adverse events because of possible drug interactions, or potentiation of side effects**					
**Ref.**	**Age**	**Sex**	**Exposure**	**Adverse event**	**Comment**
[127]	41	F	E + fish oil	Extreme ecchymoses	Improvement after discontinuation of fish oil
**Deterioration of preexisting disorder**					
**Ref.**	**Age**	**Sex**	**Exposure**	**Adverse event**	**Comment**
[128]	45	F	F	Raynaud’s phenomenon	At 6-month follow-up, reported frequent and more severe RP (the thumb was not involved) including mild digital ulcers (which had healed by the time of the visit) for about 1 month after receiving galcanezumab.
[128]	65	M	G	Raynaud’s phenomenon	Onset few weeks after fremanezumab injection, frequent episodes of RP involving all the fingers of both hands in cool temperatures
[130]	39	F	E	Paralytic ileus in a patient after undergone abdominal surgery	Paralytic ileus is a known complication of abdominal surgery
[146]	35	F	E	Xerostomia	Previous xerostomia, and patient was on amitriptyline

Abbreviations: E erenumab, F fremanezumab, G galcanezumab.

## Data Availability

Not applicable.

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
