# Peer review of "Monoclonal Antibodies against Calcitonin Gene-Related Peptide for Migraine Prophylaxis: A Systematic Review of Real-World Data"

_cells, 2022, doi:10.3390/cells12010143_

Round 1

Reviewer 1 Report

The authors have prepared a very thorough and comprehensive review of real world studies on migraine therapy. The presentation is plausible and correct. It would be desirable if the individual drugs were compared as far as possible. For example, are there differences in the parameters that have been analyzed. If this is not possible, this should be described.

Author Response

Dear Reviewer 1,

Thank you for your helpful comment and contribution.

Throughout the review, emphasis is given to the heterogeneity of the data published. As such, head-to-head comparisons were not feasible, as the published results cannot be standardized (i.e. reported means with medians). This is mentioned in the summary section (page 13).

Reviewer 2 Report

In their manuscript, Pavelic et al. performed a systematic review of real-world outcomes for anti-CGRP-monoclonal antibodies (mAbs) using the PRISMA guidelines. The authors concluded that the majority of papers support good effectiveness and tolerability of anti-CGRP-mAbs in the real-world setting.

The topic is of interest and the manuscript is well-motivated and of appropriate length. The rigor with which the meta-analysis is composed is laudable.

To improve the manuscript, I have some suggestions:

1.     Methods.

a)     The main concern is about the date of the last search for the review (april 2022, 14; more than 6 months ago). In the last months, there are some other papers reporting further novel real-word data about anti-CGRP mAbs. An updated revision is needed.

2.     Results.

a)     For the case reports, in the section of deterioration of preexisting disorder, some clinical cases reporting a worsening of preexisting inflammatory complications are lacking.

b)    Furthermore, there are some reports in real-word about the cardiovascular safety of anti-CGRP-mAbs. Recently, a myocardial infarction in a patient with migraine and triptan overuse treated with anti-CGRP receptor monoclonal antibody was reported (Cetta I, 2022).

c)     Further tables summarizing the findings could facilitate the reading (in particular for 5.4 effectiveness).

Minor comments:

- please, provide abbreviations the first time they are used in the text (e.g. RP).

- Table 1: second line is in bold; the diagnosis was made according ICHD-3 (as reported in the abbreviation) and not according ICD-10.

- There are some typos through the text.

Author Response

Dear Reviewer 2,

Thank you for your valuable and helpful comments.

In their manuscript, Pavelic et al. performed a systematic review of real-world outcomes for anti-CGRP-monoclonal antibodies (mAbs) using the PRISMA guidelines. The authors concluded that the majority of papers support good effectiveness and tolerability of anti-CGRP-mAbs in the real-world setting.

The topic is of interest and the manuscript is well-motivated and of appropriate length. The rigor with which the meta-analysis is composed is laudable.

To improve the manuscript, I have some suggestions:

  1. Methods.
  2. a) The main concern is about the date of the last search for the review (april 2022, 14; more than 6 months ago). In the last months, there are some other papers reporting further novel real-word data about anti-CGRP mAbs. An updated revision is needed.

The case reports have been updated and added where appropriate (table or in text body).

  1. Results.
  2. a) For the case reports, in the section of deterioration of preexisting disorder, some clinical cases reporting a worsening of preexisting inflammatory complications are lacking.

We added these case reports on worsening of preexisting inflammatory disease. (Page 10)

  1. b) Furthermore, there are some reports in real-word about the cardiovascular safety of anti-CGRP-mAbs. Recently, a myocardial infarction in a patient with migraine and triptan overuse treated with anti-CGRP receptor monoclonal antibody was reported (Cetta I, 2022).

We added these reports, including the case published by Cetta 2022. (Page 12)

  1. c) Further tables summarizing the findings could facilitate the reading (in particular for 5.4 effectiveness)

To visually support sections of the manuscript, such as 5.4, a visual aid in the form of supplemental tables was actually submitted, with the original manuscript. These too have been updated to reflect the latest search.

Minor comments:

- please, provide abbreviations the first time they are used in the text (e.g. RP)

All acronyms are preceded by their full versions.

- Table 1: second line is in bold; the diagnosis was made according ICHD-3 (as reported in the abbreviation) and not according ICD-10.

Table 1 has been corrected, as suggested.

- There are some typos through the text.

Typos have been identified and corrected.